# An Investigation on the Enhanced Wear Behavior of Ultrasonically Stirred Cast A356/SiO$_2$np Nano-composites

Ahmad Ghahremani [1], Amir Abdullah [1,*], Alireza Fallahi Arezoodar [1,*] and Manoj Gupta [2,*]

1   Mechanical Engineering Department, Amirkabir University of Technology, Hafez Av., Tehran 15875-4413, Iran; a.qahremani@aut.ac.ir
2   Department of Mechanical Engineering, National University of Singapore, 9 Engineering Drive 1, Singapore 117576, Singapore
*   Correspondence: amirah@aut.ac.ir (A.A.); a.fallahi@aut.ac.ir (A.F.A.); mpegm@nus.edu.sg (M.G.); Tel.: +98-21-64543419 (A.A.); +98-21-64543453 (A.F.A.); +65-6516-635 (M.G.)

**Abstract:** Metal matrix nanocomposites (MMNCs) are becoming the materials of choice in a variety of engineering and medical applications owing to their exhibiting a superior combination of targeted properties. Amongst different MMNCs, aluminum-based composites are of special importance. In many applications, a relatively inferior wear property limits the use of this valued metal in practice. However, reinforcing aluminum and its alloys by ceramics, carbon allotropes, etc., may circumvent these limitations to a great extent. In the present study, aluminum alloy A356/SiO$_2$ nanocomposite is fabricated by a vibration-assisted casting process, wherein varied amount of nanosilica, namely, 0.125, 0.25, and 0.375 wt.%, have been added to the melt. The use of power ultrasonic treatment had a great influence on the microstructure, hardness, and wear properties. Microstructural and XRD analyses were performed on the fabricated monolithic and composite samples. To evaluate wear behavior, a hardness test and pin-on-disk experiment were conducted on the samples under 60, 80, and 100 N forces at a constant speed of 1 m/s and the sliding distance was varied from 1000 to 2000 m. The abraded surfaces, wear debris, and EDS analysis were used to identify wear mechanisms. The samples having 0.125 wt.% exhibited the highest increase in hardness and the highest reduction in both friction coefficient and wear rate by 52%, 50%, and 68%, respectively. The main governing wear mechanism was abrasion, with limited evidence of delamination.

**Keywords:** ultrasonic treatment; wear; metal matrix composite; dispersion; reinforcement

## 1. Introduction

Metal matrix composites are some of the most suitable candidates for use in critical applications such as in automotive and aerospace industries. The growing need to light-weight materials for energy saving is a top priority of these industries [1,2]. Due to low density and specific strength, aluminum and magnesium are the main options for the base material in composite manufacturing [3,4].

Hard ceramics or carbon-based materials such as carbon nanotubes (CNTs) and graphite are used as reinforcement. The use of reinforcement mainly increases the strength and hardness values [5]. According to the literature, the size of the particles may also influence the mechanical properties. Compared to micron-size particles, nano-sized particles are preferred due to their unique properties and capabilities. If liquid state methods are used for processing, the particles may act as heterogeneous nucleation sites during solidification, thus refining the microstructure [6–8].

A variety of methods have been examined to produce the MMNCs, but the most common ones are casting and powder metallurgy. Casting is usually considered to be one of the most popular methods due to its low cost, high production capacity, simplicity and availability [9]. In addition to the mentioned features, there exist challenges to the production of cast metal matrix nanocomposite. The most challenging issue is the great

tendency of the nanoparticles to agglomerate due to interparticle forces such as van der Waals and the lack of proper wetting between a given melt and reinforcing nanoparticles. Theoretically, a number of strategies have been presented to deal with the poor wettability, among which reducing surface tension, increasing the number of particles, reducing particle size, boosting solidification speed, and coating particles with materials having higher Hamaker constants can be mentioned [10]. In terms of wettability, the use of materials such as magnesium [11], titanium [12,13], calcium [14], etc., all increasing the surface energy of solid particles and reducing the surface tension, may improve wettability in the interfacial region of a solid–liquid composite system. Further, the melt temperature and wetting time are also effective parameters influencing the wetting behavior of particles by the liquid metal/alloy [15].

During processing, the chemical composition of material can be affected when wetting agents are added, and, accordingly, in some applications, in order to improve the wetting condition, one may not be allowed to use a reactive agent. Further, using reactive agents may produce unwanted compounds in the manufactured composite, adversely affecting the mechanical properties [14–16]. One of the other processes that is of great interest in increasing the dispersion quality of particles, as well as boosting the affinity between a given melt and the reinforcing solid particle, is mechanical stirring [17]; in particular, high-power ultrasonic waves are used to stir the composite slurry [18,19]. In vibration-assisted casting, a stirrer with a material resistant against melt attack or a metallic stirrer coated by a resistant ceramic material is used. The main parameters of mechanical stirring are the geometry and size of the blades, the depth where the stirrer is placed, and the stirring speed and time [16,20–26]. For example, Karbalaei et al. [17] investigated the effect of stirring time on the wear resistance of stir cast $A356/Al_2O_3$ composite and found that long stirring time is useful in terms of applied shear forces; however, with respect to gas absorption and thus to increasing porosities, longer stirring may have a negative effect. They concluded that stirring has an optimal time value. The use of high-power ultrasonic waves is considered as a mechanical stirring process, wherein cavitation is responsible for deagglomerating the particles and dispersing them uniformly throughout the host melt matrix, bringing opportunities for heterogeneous nucleation [18,27–29]. In addition to the deagglomeration and uniform distribution of particles in the melt [29,30], other phenomena such as degassing [31], increasing the surface energy of solid particles [18], cleaning surfaces from contaminants [32,33], preventing segregation [34], the reduction of thermal stresses [35], the uniform distribution of different phases in the bulk [36] and the creation of a strong acoustic flow in the melt, mixing melt slurries [37–39], have been mentioned in the literature. Mechanical methods, especially the effective use of ultrasonic waves, are more efficient and less problematic compared to other methods, especially for realizing a fine and uniform distribution of nanoparticles in a given cast MMNCs [40].

In this study, A356 alloy has been chosen due to its high castability [41]. Moreover, as mentioned earlier, adding elements such as Mg, Ti, Ca, and Zr causes better wettability in metal–ceramic nanocomposites [42,43], and these elements are mainly present in A356 composition. For strengthening, nanosilica is used as reinforcement, being relatively inexpensive and available, and, as far as we know, less research has been conducted on it. Compared to SiC, nanosilica has a lower contact angle with molten aluminum, as indicated by Tekmen et al. and Hashim et al., who tried to oxidize the SiC surface by thermal treatment to create a better wettable $SiO_2$ layer on the silicon carbide surface [32,42]. Ultrasonic-assisted casting is used in this work to uniformly distribute silica nanoparticles in A356 melt. Since little is known of the mechanical and tribological behavior of the $A356/SiO_2$ nanocomposite, the present paper primary addresses the wear response of the ultrasonically stirred and cast $A356/SiO_2$ composites.

## 2. Materials and Methods

A356 aluminum alloy is used as the base matrix alloy, being supplied from the Materials Engineering Department at Isfahan University of Technology, Isfahan, Iran. The

chemical composition of the alloy is given in Table 1. In each casting experiment, 1600 g of A356 alloy was poured into a graphite crucible to be melted under the protection of argon gas with a purity of 99.99%. Upon raising the melting temperature to 750 °C, 98.5% purity silica nanoparticles, supplied by Fadak-Isfahan New Technologies Complex, with the specifications presented in Table 2, were added to the melt (0.0, 2.0, 4.0, and 6.0 g). The corresponding amounts of the added reinforcing agent are, respectively, 0.0%, 0.125%, 0.25%, and 0.375%. The mentioned samples are named, respectively, AMNC0, AMNC2, AMNC4, and AMNC6 for ease of review. Figure 1 shows TEM and SEM images of the used silica nanoparticles.

**Table 1.** Chemical composition of A356 aluminum alloy.

| Zn | Mn | Cu | Fe | Ti | Mg | Si | Al | Material |
|---|---|---|---|---|---|---|---|---|
| 0.03 | 0.05 | 0.1 | 0.15 | 0.2 | 0.3 | 7.5 | Balance | wt.% |

**Table 2.** Specifications of nanosilica reinforcement.

| Value | Unit | Properties and Test Method |
|---|---|---|
| 210–240 | $m^2/g$ | Specific surface area (CTAB adsorption) |
| 30–50 | nm | Main particle size (TEM) |
| 2.65 | $g/cm^3$ | Density |
| 0.13–0.15 | $g/cm^3$ | Tamped density ISO-787-11 |
| 1.3 | $Wm^{-1}\,K$ | Thermal conductivity |
| 12.3 | $10^{-6}\,K^{-1}$ | Thermal expansion coeff. |
| 1830 | °C | Melting point |
| $\leq 98.5$ | % | $SiO_2$ Content ISO 3262/17 |

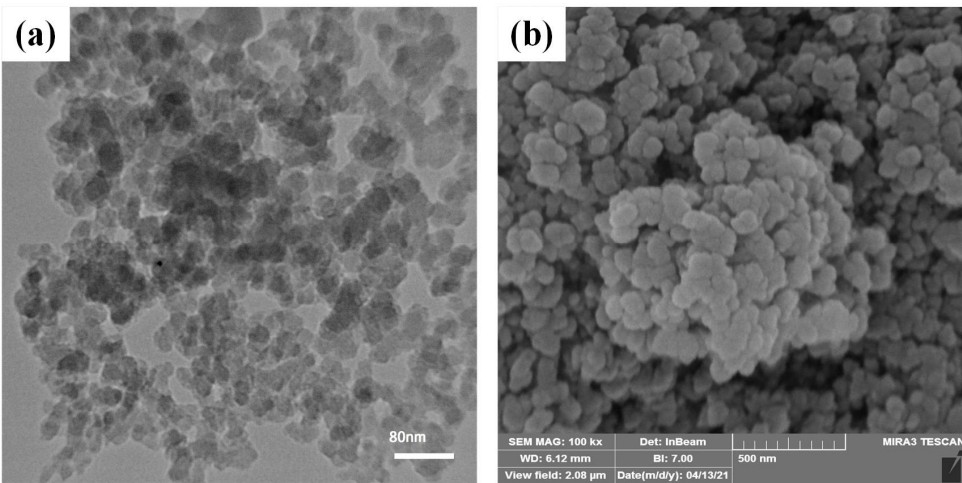

**Figure 1.** (**a**) TEM and (**b**) SEM of the nanosilica reinforcement used in this study.

The A356/SiO$_2$ composite samples were fabricated by ultrasonic-assisted casting, wherein a high power 2.0 kW piezoelectric transducer with a resonant frequency of 20 kHz was used. The transducer was first designed along with its vibration horn. After simulating and determining the exact geometry, the horn concentrating the ultrasonic vibrations was fabricated. There were six ring piezoelectrics that were tightened by a steel bolt between the steel backing part and the titanium matching part. The ultrasonic horn made of a Ti6Al4V alloy with specific geometry and dimensions was connected to the transducer in part of the vibrating node (see Figure 2a). The transducer with such configurations can also be used in some other manufacturing processes, including powder consolidations, machining, welding, etc. [44–47]. To protect the transducer, it was placed inside metal housing and the concentrator was placed outside the mentioned housing. To prevent piezoelectric damage

from the generated heat, as well as from the heat transferred from the melt, the cooling system on the chamber is designed and installed exactly at the location of the piezoelectrics. In this system, cooling air is blown into the chamber and a fan was installed at the end of chamber to exhaust the air. The vibration head of the concentrator was immersed in the melt at a temperature of 750 °C and sonication was performed for 5.0 min. It should be noted that this period of time was optimized after trial and error. In other words, for times less than 5.0 min, the operation of ultrasonic waves was not effective enough, and for longer times, the probability of Ti dissolving at the end of the horn increased at high temperatures and high frequency vibration, causing contamination of the melt and unwanted effects on the microstructure. In addition to the issue of the melt being contaminated by Ti, the possibility of piezoelectrics being damaged under long-time exposure increases and can cause the operation to stop.

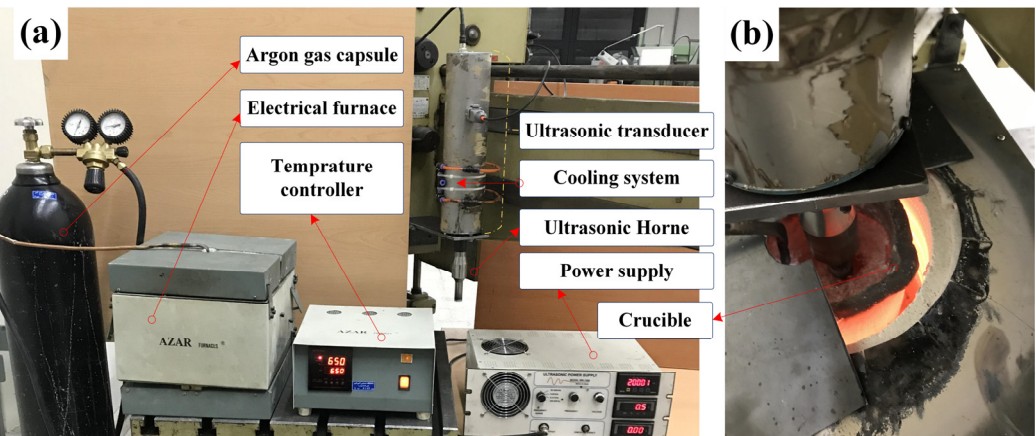

**Figure 2.** (**a**) Experimental setup and (**b**) A356/SiO$_2$ melt under sonication.

The layout of the experiment can be seen in Figure 2. A356 aluminum ingots were first placed in the graphite crucible, after accurate weighing, to be melted by an electric furnace under the protection of argon gas. Silica nanoparticles were then preheated (at 250 °C for 1 h), after accurate weighing by a scale with a resolution of 0.1 mg, so that the moisture was removed; finally, they were gradually added to the melt at the same time as applying ultrasonic vibrations at a temperature of 750 °C. After the melt was treated by ultrasonic vibrations, the composite slurry was immediately poured into a steel mold with an internal cross section of 60 × 60 mm, a wall thickness of 10 mm, and a height of 200 mm. After leaving the cast materials from the mold, the cooled ingots were machined and cut by a wire-cut machine (Wire-EDM, Tabriz Machines Co., Tabriz, Iran). They were then used to perform hardness measurements, wear tests, and microstructural analyses.

*Characterization*

Vickers hardness tester (DVKH-1, OGAWA SEIKI Co., LTD., Shinjuku-ku, Japan) with 5 kg weight and 15 s dual time was used in accordance with ASTM E92-17. To ensure repeatability, the tests were conducted with 5 repetitions per sample. Moreover, the hardness curve was drawn in the longitudinal direction of the cast ingots. To estimate the porosities of the cast samples, Archimedes test was performed.

A standard pin-on-disk apparatus, supplied by Advanced Modern Industry Company, Tehran, Iran, was used to measure the wear rates of the samples. The schematic and real images of the apparatus are shown in Figure 3. One of the special features of the mentioned device is the placement of the disc vertically; hence, almost all wear debris may not remain on the surface of the disc due to gravity and centrifugal force. In fact, in those wear experimental setups with a horizontal disc, some worn particles and hard reinforcing particles are separated but remain on the surface of the disc, and they may enter the space between the disc and the pin, leading to the scratching of the pin surface or

providing conditions for three-body abrasive wear, capable of inducing errors either in the wear results or in wear mechanisms [48]. In this device, power is supplied through a linear actuator and the values of 10 to 100 newtons can be applied. The friction force can also be measured through a load cell that is parallel to the disk surface. A Pt100 type sensor is also installed to measure the temperature.

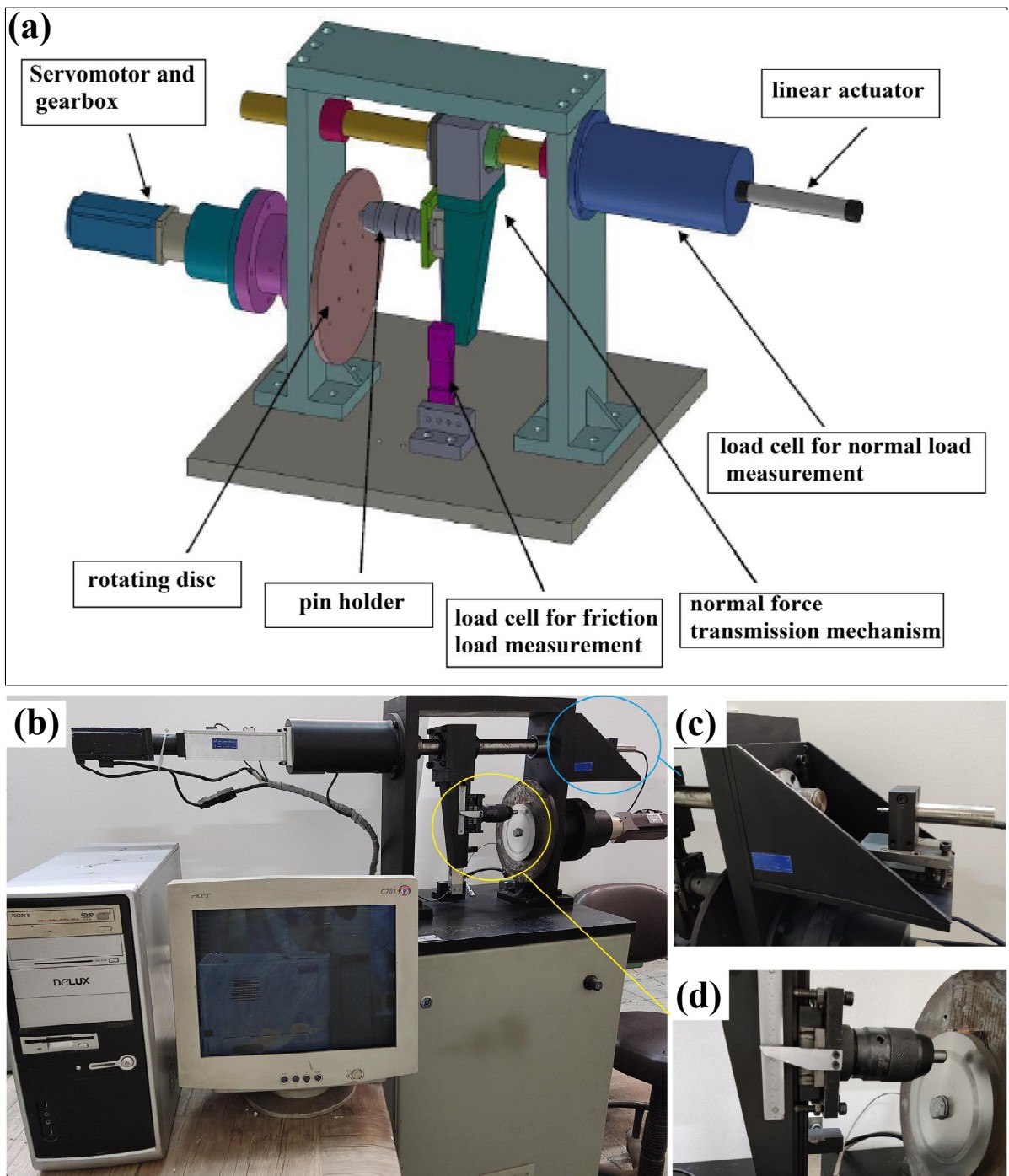

**Figure 3.** (**a**) Schematic of device with pin on a vertical disc, (**b**) actual image of the device, (**c**) wear measurement sensor, and (**d**) pin and disc in higher magnification.

The samples, in the form of pins, were prepared and gripped by a special holder. The abrasive disc was made of AISI 52,100 steel with the hardness of 65 HRC, diameter of 180 mm, thickness of 15 mm, and the surface roughness of 3 μ. This disk rotates at a maximum speed of 300 rpm, having the ability of closed loop control of an accuracy

of 0.01%. The disc is perfectly perpendicular to the pin. A sensor is installed in this device to show the wear volume based on the amount of wear of cylindrical samples with micron accuracy. The samples were prepared in the form of pins with dimensions of 8 mm in diameter and 25 mm in height. To obtain accurate results, the end of each pin was spherically machined and polished with SiC sandpapers from 240 to 3000 grades (see Figure 4).

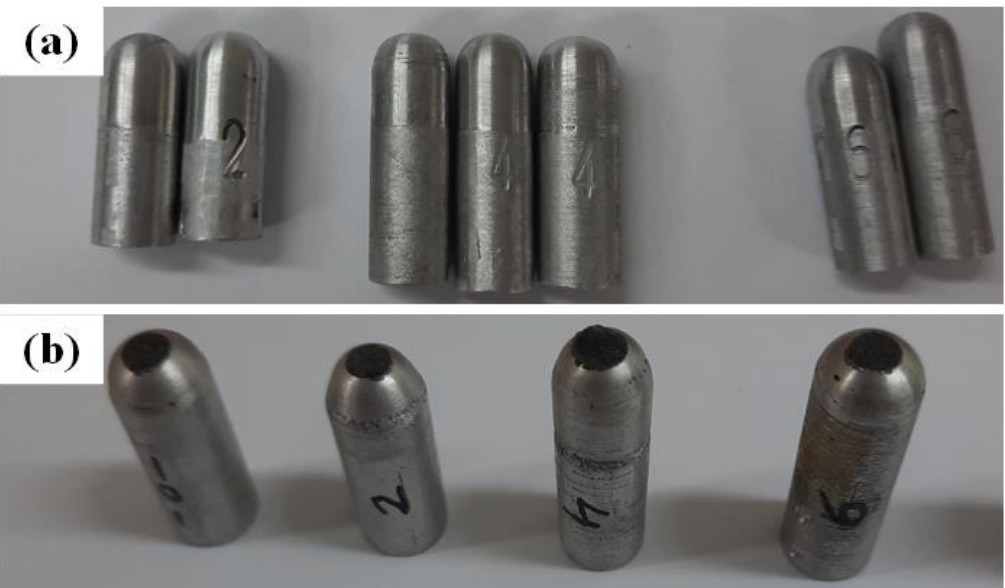

**Figure 4.** Fabricated samples: (**a**) before wear test; (**b**) after wear test.

The samples were first cleaned with a soft cloth just before and after the wear tests. They were then washed with acetone to remove contaminants and dried using cold air. Finally, they were weighed with a resolution of 0.1 mg. The weight difference was analyzed and studied as the main indication of wear values. The wear test parameters are given in Table 3.

**Table 3.** The parameters used in wear experiments.

| Distance (m) | Speed (m/s) | Normal Force (N) | Nanosilica (g) | Parameter |
|---|---|---|---|---|
| 1000, 1500, 2000 | 1 | 60, 80, 100 | 0, 2, 4, 6 | Value |

In order to analyze the microstructure of the cast samples, metallographic photographs were taken by an Olympus PME-3 optical microscope (LECO Co., Michigan, MI, USA) at a magnification of 100×. SEM images were also taken from the primary samples to check the microstructure for abraded surfaces and to estimate the wear mechanisms by FE-SEM MIRA3, TESCAN Co. (Brno, Czech Republic). In order to further investigate the effect of silica reinforcements, XRD analysis was performed using Inel, EQUINOX 3000, France, working with a voltage of about 40 kV and a current of 30 mA. The wavelength and the range of the mentioned XRD device are, respectively, $\lambda = 0.154$ nm and $2\theta = 0–110°$, with a step size of $0.01°$.

### 3. Results and Discussion

*3.1. Microstructural Evaluations*

After machining, the cast ingots were cut by a wire-EDM to prepare the standard samples with desired dimensions. These samples were ground using 220- to 3000-grade emery clothes and then polished with a polishing machine using 0.3-micron alumina suspension. Upon washing and drying, the specimens were etched by a Keller's agent.

Figure 5 shows the microstructure with 100× magnification of AMNC0, AMNC2, AMNC4, and AMNC6.

**Figure 5.** Optical microscopy results: (**a**) AMNC0, (**b**) AMNC2, (**c**) AMNC4, and (**d**) AMNC6.

The microstructures given in Figure 5 show that the microstructure of the A356 aluminum composite could substantially change only if 0.125 wt.% silica nanomaterial was added to the melt. Reduction of the grain size and break down of the dendritic structure is observed in this weight fraction, owing to the application of powerful and effective ultrasonic irradiation in deagglomerating and dispersing the silica particles evenly throughout the melt slurry. The uniform dispersion and distribution of nanoparticles may considerably facilitate heterogeneous nucleation, resulting in a finer microstructure. Cavitation being one of the main consequences of ultrasonic treatment may break the dendritic arms and structures. By adding more nanoparticles to the melt in the AMNC4 nanocomposite, no improvement was seen in the microstructure (Figure 5c); a coarser structure and the dendritic structure appeared. In fact, by adding nanoparticles twice, the viscosity of the melt increased, and as a result, the ability of ultrasonic dispersion decreased slightly, and the structure become coarser to some extent. For the same reason, by adding more nanoparticles, more dendritic and coarser structures are seen, owing to a decrease of ultrasonic efficiency in more viscous composite slurry (Figure 5d). With this, it can be seen that the optimal percentage of nanoparticles in these conditions is 0.125 wt.%.

*3.2. XRD Results*

A comparison of the XRD patterns of the different samples is depicted in Figure 6. Since the percentage of the added nanoparticles was very low, and since the nanoparticles were very small and fine, no peak related to silica was observed in any of the XRD patterns. Hosseini and his colleagues have also reported the absence of peaks related to the reinforcing phase, attributing it to the small size of the nanoparticles and the lower weight fraction of the reinforcing particles [49].

The greater broadening and lower intensity of the peak of the 0.125 wt.% confirm that the greatest reduction was achieved due to the addition of nanoparticles in the AMNC2 sample. The addition of reinforcements may cause the distortion in the structure of the nanocomposite, and the result is a broadening of the XRD pattern of the nanocomposite compared to the monolithic alloy [50]. Based on an XRD analysis, Sharma et al. [51] showed that the pattern of XRD peaks in the Al6082/SiC nanocomposite shift a little towards

2θ angles—lower than in the un-reinforced alloy. Similarly, the shifts towards lower angles (left side) are seen when $SiO_2$ is added. This pattern has also been reported by others in the literature [50].

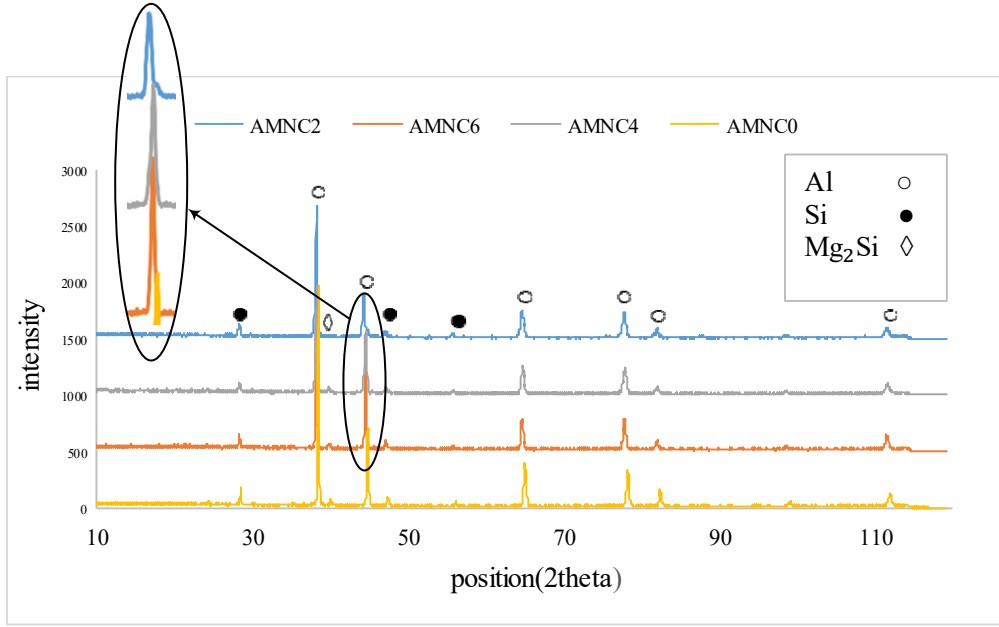

**Figure 6.** X-ray diffraction patterns of the A356 alloy and Al356/$SiO_2$ nanocomposites with 0.125, 0.25, and 0.375 wt.% nanosilica.

### 3.3. Hardness Results

To determine the hardness changes along the length of the cast sample, a hardness test was performed at each 2 cm interval. After sectioning the surface into three separate zones, i.e., the perimeter, middle points, and the center of the piece, the hardness test was performed, and the test was repeated at least five times in each zone. The results are used to obtain the hardness profile along the length of the samples. Figure 7 shows the hardness profile of each monolithic and composite sample.

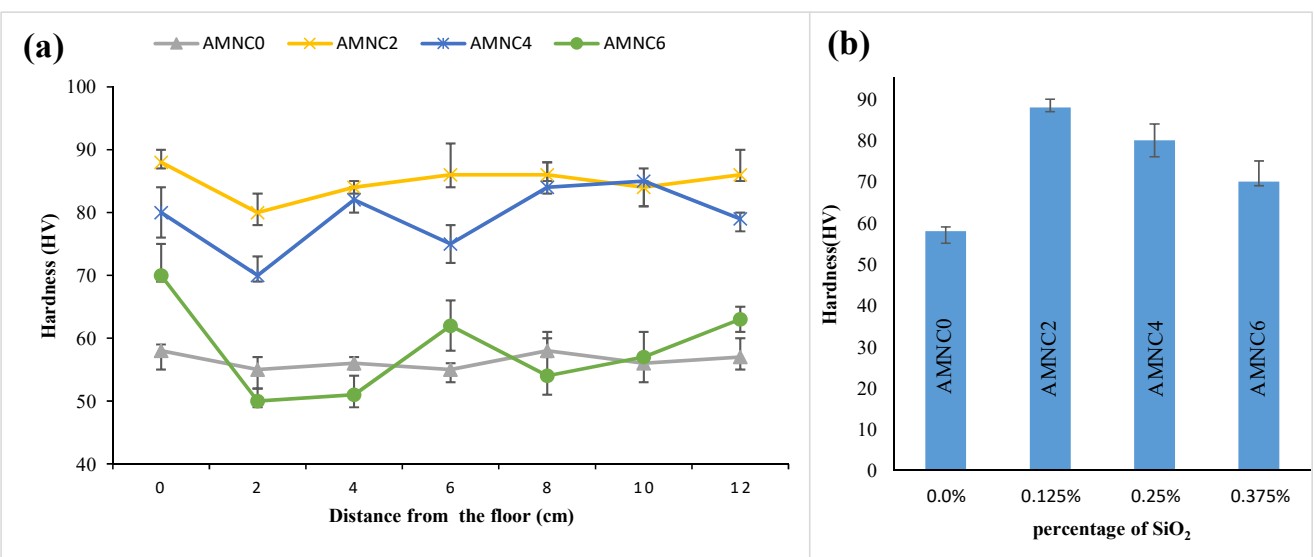

**Figure 7.** (**a**) Hardness profile along the length of the cast ingot; (**b**) comparison of samples AMNC0, AMNC2, AMNC4, and AMNC6 prepared from the bottom of the cast ingot.

As seen in Figure 7, the composites with different weight fractions showed higher hardness values than the base metal. The highest hardness value of 88 HV was seen in the AMNC2 nanocomposite, but the value decreased with an increase in silica content, reaching to the lowest amount of 68 HV in the AMNC6 sample. The microstructure shown in Figure 5 also confirms that the structure of the AMNC2 sample is quite fine. The fine structure based on the Hall–Patch relationship contributes to the greater strength of the nanocomposite [52]. On the other hand, the added particles can increase the strength of the nanocomposite through other mechanisms such as Orowan, CTE (coefficient of thermal expansion), and EM (elastic modulus) [35]. With an increase in the wt.% of reinforcement, the viscosity of the aluminum melt increases, and as a result, ultrasonic waves become less effective than when the viscosity of the fluid is higher [27]. Moreover, with the addition of nanoparticles, porosity may have increased; therefore, it can be concluded that gradually, with the addition of nanosilica, the degree of nanosilica dispersion may tend to decrease and the severity of porosity increases. Others have also reported that by adding relatively small percentages of ceramic nanoparticles, the hardness shows a significant increase, and with further addition of reinforcing phase, the increasing trend slowed down or had a decreasing trend [52–54].

The noteworthy point in the hardness profile of Figure 7a is that the hardness is higher ate the bottom and top of the cast samples, and there are fluctuations along the length of the sample, which mainly depend on the cooling conditions and the weight fraction of nanoparticles in that area. The reason for the higher hardness in the bottom can be due to the following reasons: (i) The bottom is the first place where the melt is in contact with the mold; hence, the cooling speed in the bottom is higher than any other points; it means faster cooling leads to finer structure and thus higher hardness of the samples. In this project, the steal mold is cooled by water from its bottom part, which contributes to the higher temperature gradient in this part. (ii) One of the mechanisms of engulfment of small particles in the molten metal, based on the capture theory, is the viscous capture mechanism, stating that if the cooling rate is higher than a critical rate, the possibility of particle engulfment at the solidification front will be higher [10]. Considering the mentioned theory, more particles were probably engulfed in this area and harder nanocomposites formed in this area. In the case of greater hardness in the upper parts, it is also possible to raise the possibility that the solidification front is from the lower part of the mold, that the nanoparticles are possibly rejected as impurities, and that the upper part will be richer from nanoparticles than the middle parts, and the observed difference in hardness is due to the higher amount of nanoparticles at the top of the solidified sample.

The Archimedes test was used to measure the density of the cast samples. The test is based on weighing a sample in two different fluids such as air and distilled water. Generally, the reference fluid is air, and the second fluid is distilled water. Each test was repeated three times; each sample was carefully washed and dried with washing liquid and acetone before testing. We obtain the density of the sample ($\rho_s$) based on equation 1, wherein $m_{air}$ is the weight of the sample in air, $m_{dw}$ is the weight of the sample in distilled water in the immersion state, and $\rho_{dw}$ is the density of distilled water. The air density is neglected.

$$\rho_s = \rho_{dw} \times \frac{m_{air}}{(m_{air} - m_{dw})} \tag{1}$$

The Archimedes test was performed for the monolithic sample and nanocomposites with different nanosilica weight fractions; the results are shown in Table 4.

**Table 4.** Porosity percentage of monolithic and nanocomposite samples, i.e., AMNC2, AMNC4, and AMNC6.

| AMNC6 | AMNC4 | AMNC2 | AMNC0 | Samples |
|---|---|---|---|---|
| 2.5 | 1.6 | 1.2 | 0.6 | Porosity(%) |

### 3.4. Wear Response

Researchers have reported that the use of hard ceramic reinforcements—especially those in nanometric scales—in aluminum matrix nanocomposites may protect the soft matrix phase against wear, reducing the severe surface shear strains typically observed in the unreinforced samples. Hard ceramic particles may act as an abrasive member and load bearer. The microstructure of the base alloy and its history, such as the manufacturing route, as well as mechanical and heat treatments, may have a great influence on the wear resistance of its composite material. This effect is mainly through the microstructural change, the dispersion of reinforcing particles, voids and porosities, the bonding in reinforcement–matrix interfacial region, and the mechanical properties [5,55].

A common criterion, i.e., weight loss, was used to compare and study of the wear of samples. Figure 8 demonstrates the changes in the weight loss of the samples under different conditions.

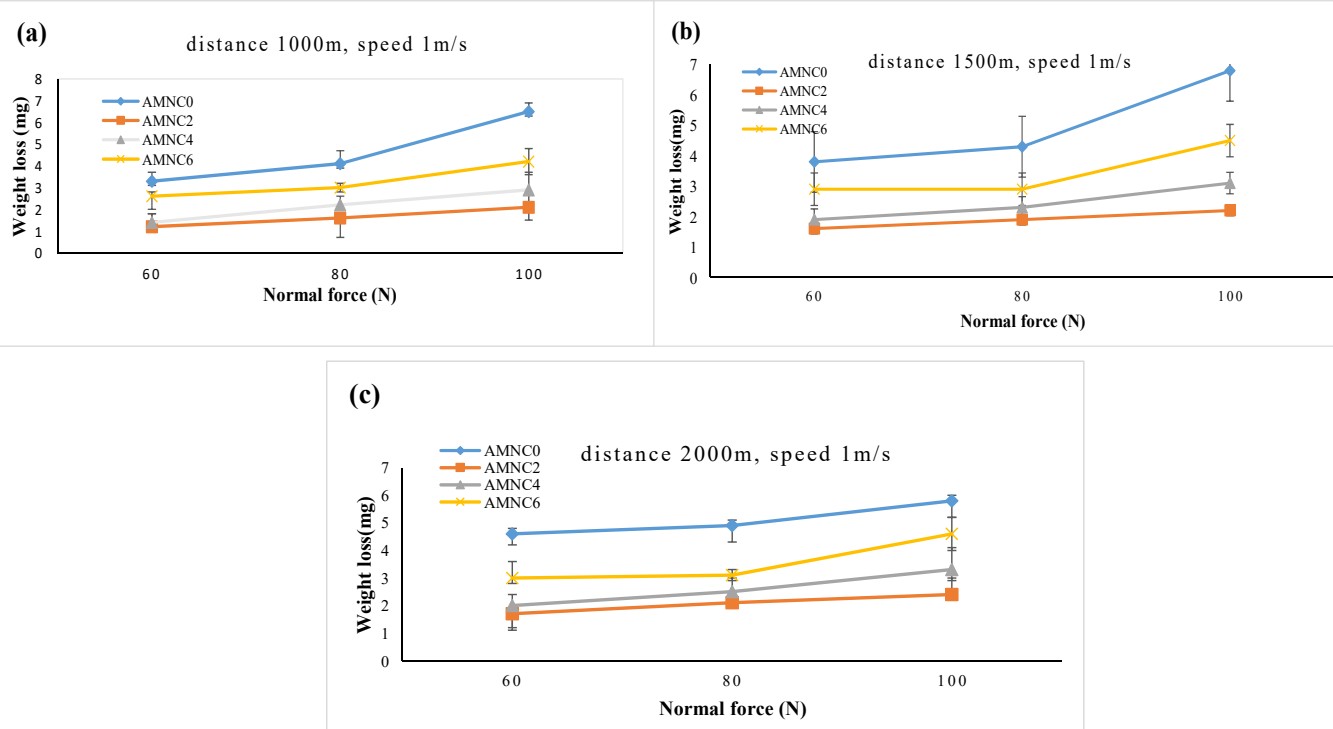

**Figure 8.** Weight loss comparison versus the vertical force of wear at a speed of 1 m/s and the distance of (**a**) 1000 m, (**b**) 1500 m, and (**c**) 2000 m.

To obtain the wear rates and for a better comparison between the composite and monolithic samples, Equation (2) is used.

$$Wear\ rate = \frac{Weight\ loss \times 10^{-3}}{\rho \cdot D} \tag{2}$$

In Equation (2), wear rate is in mm$^3$/m, and weight loss is in g; $\rho$ is the actual density of a given sample being measured by the Archimedes test in g/cm$^3$, and $D$ is the distance traveled in meters.

The wear rate in terms of force in different distances of 1000, 1500, and 2000 m is presented in Figure 9.

The wear rate can change according to Archard relationship and the hardness of the samples [56–58]. According to the hardness result, the addition of up to 0.125 wt.% nanosilica may considerably increase the hardness, but it tends to gradually decrease when further reinforcement is added to the aluminum melt. The wear rate also follows the same

hardness trend. There is a significant difference between the wear rate of the AMNC2 sample and that of the A356 monolithic alloy. For example, the wear rate at a distance of 1500 m and a force of 100 N has decreased from about 0.00168 mm$^3$/m in the unreinforced alloy to 0.00054 mm$^3$/m in the AMNC2 nanocomposite, attaining more than three times the reduction in the wear rate. Despite the fact that the porosity of AMNC2 is slightly higher than the monolithic material, and that the porosity usually has a negative effect on the wear resistance, the extreme reinforcement-induced hardness and microstructural refinement in the AMNC2 sample may finally lead to a significant decrease in the wear rate. Due to lower viscosity, it seems that ultrasonic irradiation may treat more effectively the base alloy melt or the nanocomposite melts with lower silica weight fractions. The ultrasonic treatment is able to create a more homogeneous structure through the cavitation mechanism, wherein the structure can be considerably refined, and those clustering and agglomerations can be effectively disintegrated and finely dispersed in the matrix. In addition to deagglomeration, ultrasonic waves can also cause degassing to some extent. The phenomenon of ultrasonic degassing causes stronger bonding between the nanoparticles and the matrix, leading to better load transfer from matrix to nanoparticles [18]. Adding nanoparticles may cause micro-void nucleation around the nanoparticles. If nanoparticles are agglomerated, there might be more porosity inside the aggregates; consequently, they prevent the flow of melt and cause the formation of cavities [17]. By applying vibrations with a frequency of 20 kHz, ultrasonic waves can mitigate the porosity originating around the nanoparticles.

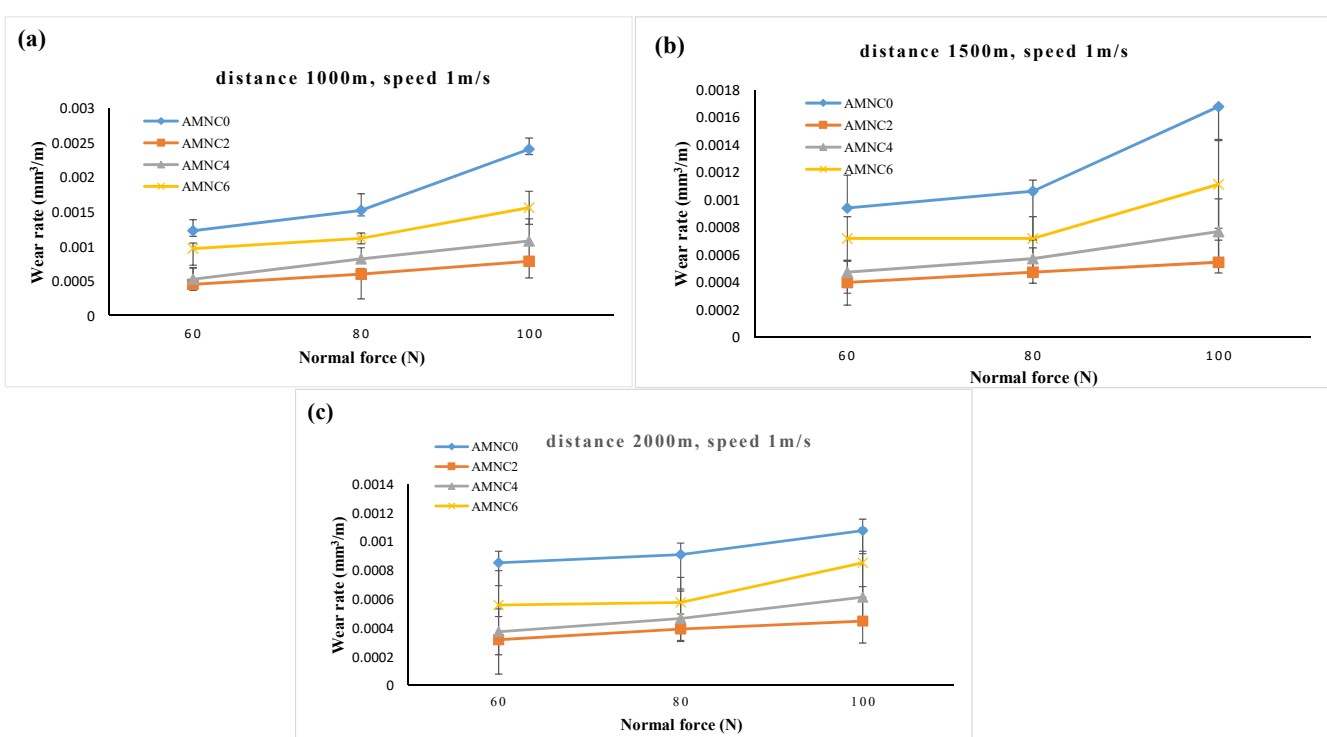

**Figure 9.** Wear rate versus the normal force at the speed of 1 m/s and a distance of (**a**) 1000 m, (**b**) 1500 m, and (**c**) 2000 m.

An increase in the normal force leads the plowing force (or friction force) to increase. In these conditions, more penetration into the material occurs and sub-surface fracturing will be expected; further, delamination may be activated when more shear strain is applied [59]. It is worth noting that the wear resistance improvement is higher at greater forces. In other words, nanoparticles have a greater effect on the load bearing capacity at higher wear forces and exhibit more wear resistance [59,60]. For example, at a distance of 1500 m and a force of 60 N, a 57% reduction in the wear rate is seen in AMNC2 compared to the unreinforced

alloy, while the wear reduction rate under the same conditions and a force of 100 N can be up to 68%.

### 3.5. Coefficient of Friction

Figure 10 shows the coefficient of friction (COF) for the unreinforced alloy and the nanocomposites of AMNC2, AMNC4, AMNC6 in terms of distance 1000 m and under 60 N force. In the first stages, because the surfaces are not yet fully involved, lower values of the friction coefficient are observed, but then it increases quickly, and after this stage, a large discrepancy is seen in the values. The average values are depicted by the non-continuous line. The average value of the coefficient of friction in terms of the nanosilica weight fraction is shown in Figure 11a. An important point is that all the composite samples have a lower friction coefficient than the unreinforced sample. The COF of the monolithic sample is about 0.52, and it reached 0.27 in the AMNC2 nanocomposite, attaining a 50% reduction. The non-continuous line that indicates the average amount of COF of the AMNC2 sample is increasing. It is possible that with increasing distance and temperature due to the low weight fraction, the resistance to tangential forces has decreased and the friction coefficient has increased slightly. By increasing the weight fraction in AMNC4 and AMNC6 samples, the friction coefficient increased slightly. The COF itself is affected by the sum of the factors that are given in a simple form of the following relationship [61].

$$\mu = \mu_{el} + \mu_{pl} + \mu_{ad} + \mu_d \tag{3}$$

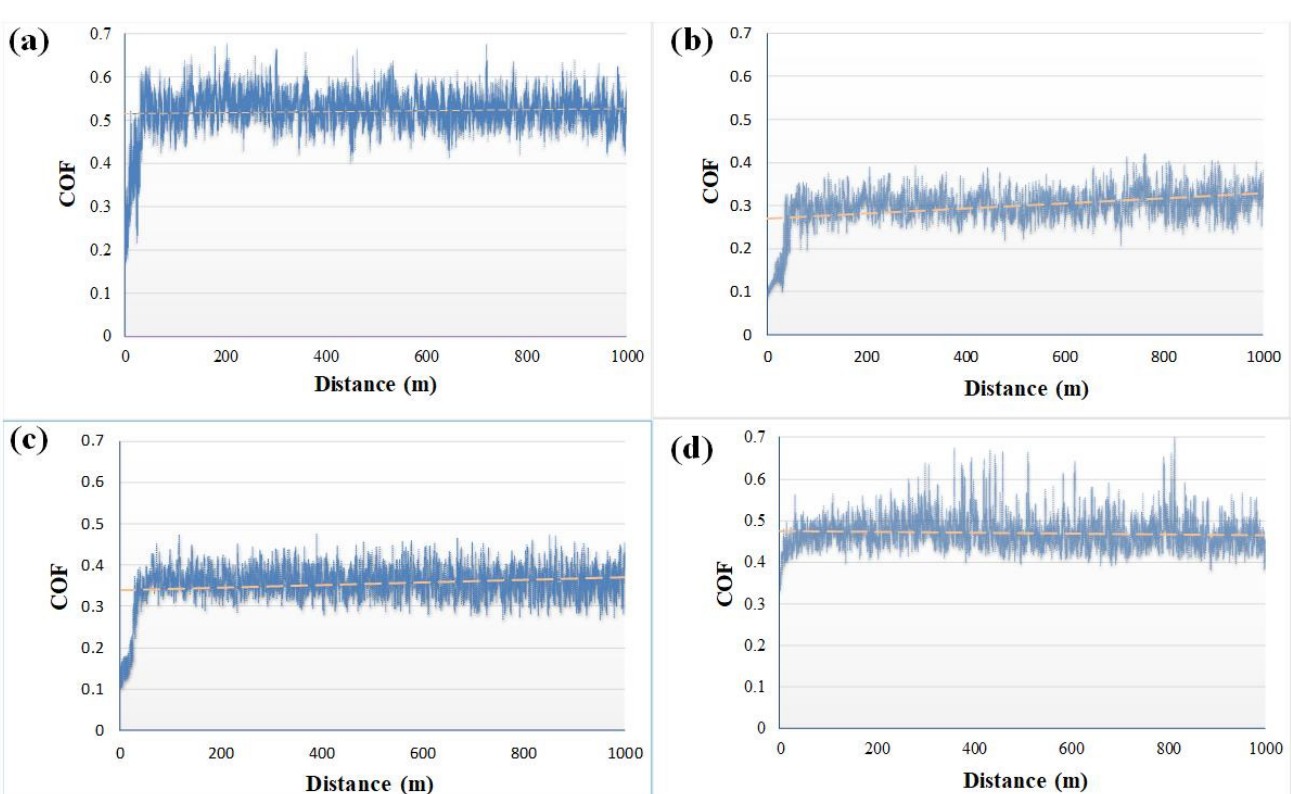

**Figure 10.** COF versus by the distance for: (**a**) AMNC0, (**b**) AMNC, (**c**) AMNC4, and (**d**) AMNC6. The average values of COF depicted by dotted line.

In this regard, $\mu_{el}$, $\mu_{pl}$, $\mu_{ad}$, and $\mu_d$ are, respectively, related to elastic deformation, plastic deformation, adhesion between Al particles, and the effect of silica nanoparticles. As is clear from Figure 5, the AMNC2 nanocomposite has a finer structure and according to the Hall–Patch equation; it can be inferred that the elastic deformation term becomes greater. It has also been observed that the mechanical strength of nanocomposites has

increased compared to the base alloy, as is confirmed by the hardness test results. Therefore, the term of plastic deformation is also higher in composite samples. In the meantime, increasing the hardness and toughness of the material leads to a lower contact in the mating surfaces; consequently, the adhesive forces attracting the two surfaces decrease, and it can be concluded that the value of $\mu_{ad}$ may also reduce in the composite samples [62]. Finally, according to references [63–65], the presence of $SiO_2$ particles has a lubrication property, reducing the friction coefficient between the two surfaces. Considering why the COF does not decrease with the increase in the nanosilica weight fraction from 0.125 to 0.375, it can be noted that despite the presence of more nanoparticles, the porosity also increases, and the reason is the lower efficiency of ultrasonic waves at higher viscosities. Probably, a higher porosity reduces interfacial bonding, and the mentioned clustered nanoparticles/agglomerates are separated from the matrix due to the tangential forces, and, in a way, it provides three-body wear conditions between the two surfaces. As a result, scratching the surface by the mentioned large agglomerates, the surface roughness may increase in higher weight fraction composites [17]. By increasing the nanosilica weight fraction, in addition to increasing the porosity rate, there is also the possibility of particle agglomeration, being separated from each other due to lower shear forces, and this may cause more surface destruction [66]. Therefore, in summary, we can point to factors such as increasing the porosity and agglomeration in the weight fractions higher than the sample with the optimum silica reinforcement (here 0.125% by weight), which do not allow further wear resistance improvement.

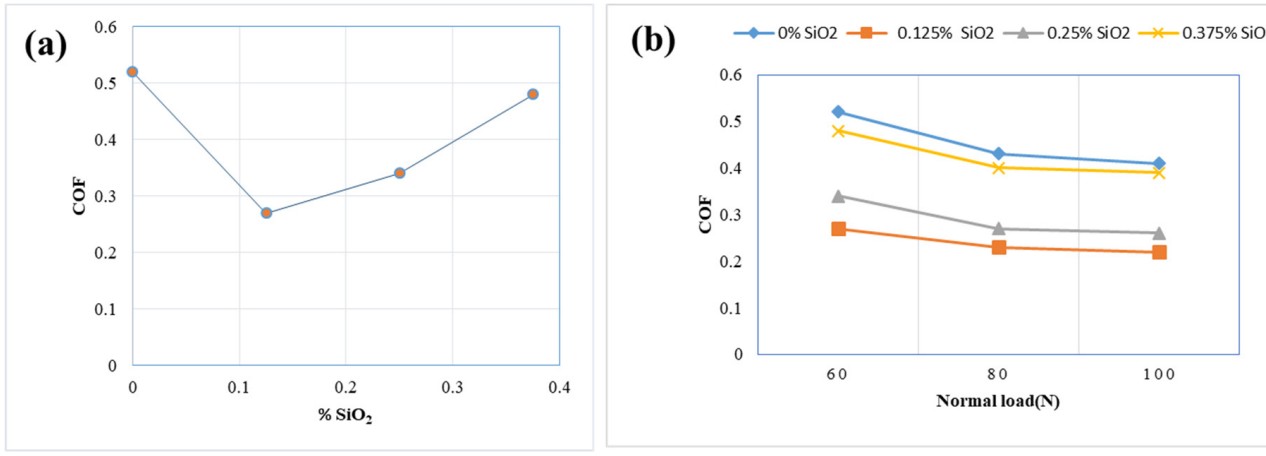

**Figure 11.** (**a**) COF versus nanosilica weight fraction; (**b**) friction coefficient versus normal force.

COF has high scattering values in all the samples, probably due to the sticking and separation of pin and disc surfaces. The oscillation range in the AMNC2 composite sample is much narrower compared to the monolithic alloy, indicating much more uniformity in the mentioned weight fraction.

Figure 11b shows the variation of COFs in terms of normal forces. It is seen that an increase in normal force results in a decrease in COF. The slope of the reduction of the COF is greater from the force of 60 to 80 N. The change of the COF in the AMNC2 sample does not considerably change, while the unreinforced sample has decreased significantly due to the wear force. Naturally, when the normal force increases, the tangential force, the elastic and plastic deformation, and the penetration depth of the asperities of the two surfaces may increase accordingly, being responsible for the rising temperatures at the contact point. As the temperature goes up, local softening occurs and the tangential force may reduce [67,68]. The reduction of the COF induced by the increased normal force has also been observed by others in the literature [50,69,70]. Karbalaei Akbari [53] has reported the mentioned phenomenon while studying the tribology of Al356/$TiO_2$ and $TiB_2$ nanocomposites, attributing it to the formation of a tribo-oxide layer as a result of the increasing force.

### 3.6. Wear Mechanisms

The SEM results of the abraded surfaces under the speed of 1 m/s, the distance of 1500 m, and the force of 80 N are shown in Figure 12. The mechanisms observed in these samples are a combination of the adhesive mechanism, delamination, and abrasion. The wear path can be recognized by an arrow. In the lower left corner of each figure is a lower magnification image demonstrating the amount of material removed by the wear test. The surface of the unreinforced alloy shows a significant degradation and there is almost no trace of the scratch mechanism. The creation of deep pits and plastic deformation, as well as delamination with large thicknesses, being the characteristics of the adhesive and delamination mechanisms, respectively, can be clearly detected. Looking at the image with lower magnification in the left and bottom corner of Figure 12a, it can be seen that the flash is created along the wear path which somehow refers to the plastic deformation of the base alloy and stretching in the wear path. When metal-to-metal contact occurs, material transfer occurs between the two surfaces in contact, and the dominant mechanism in this type of contact is mainly adhesive [71].

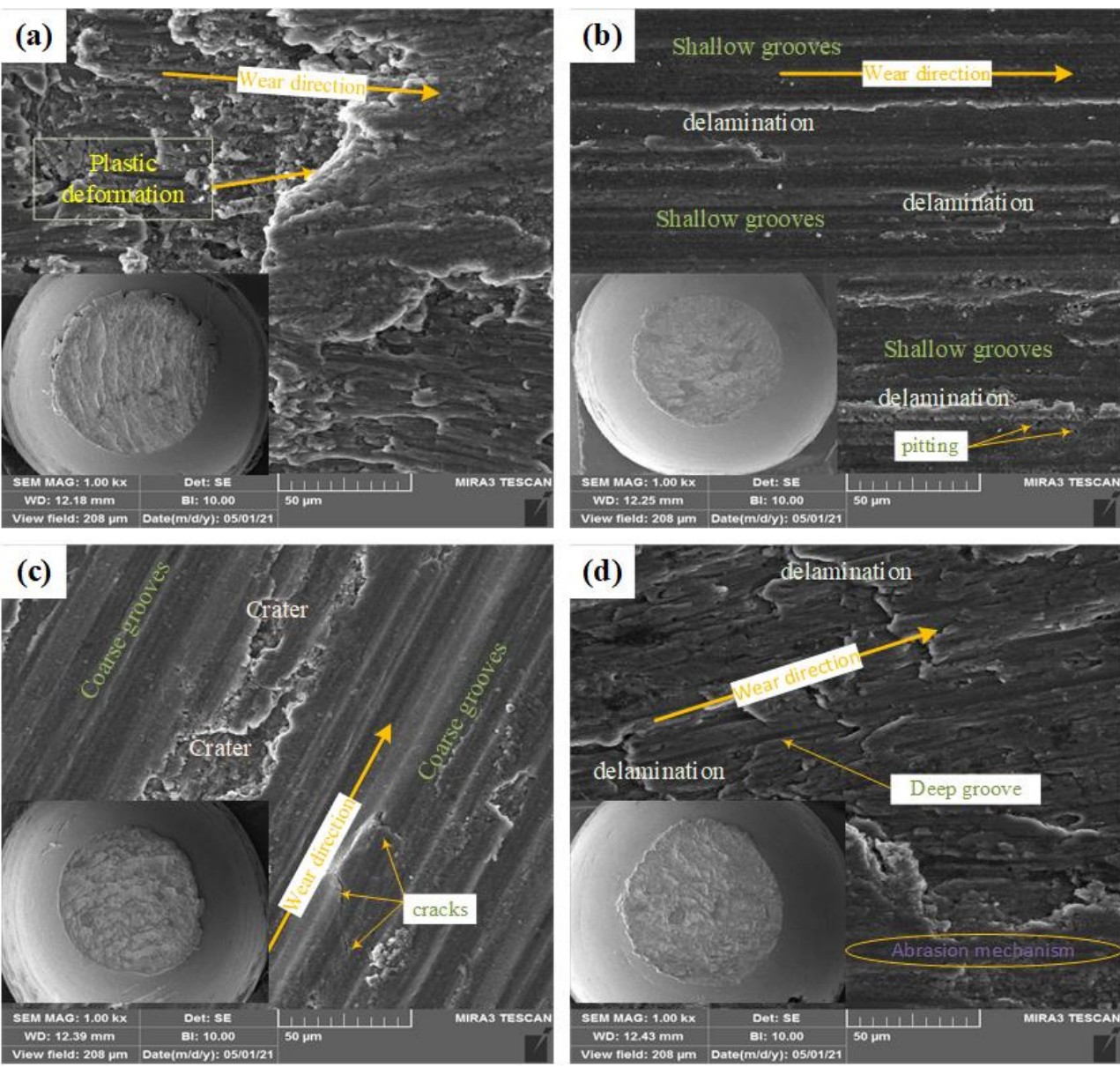

**Figure 12.** SEM images of the worn surfaces: (**a**) AMC0, (**b**) AMNC2, (**c**) AMNC4, and (**d**) AMNC6.

Figure 12b shows the worn surface of AMNC2 nanocomposite. The amount of flash generated in this sample is significantly reduced, probably owing to the hardness of the nanocomposite as well as the refined microstructure of the AMNC2 composite. Regular and relatively shallow grooves along the wear path, being the feature of scratch mechanism, are evident in this image. These grooves are created by the asperity of the opposite surface penetrating the surface of the pin and are mainly associated with the cutting and separation of ribbon-like chips from the surface. This mechanism often occurs in metal-to-metal contact when the pin surface is sufficiently hardened; however, if the subsurface layer is not strong enough, delamination is caused wherein microcracks are created which then connected each other parallel to the surface [5]. The delamination mechanism can also be seen in this sample, to some extent. To a small extent, pitting is also visible, which is a characteristic of the delamination mechanism. Therefore, the dominant mechanism in this composite is mainly abrasion and then, to some extent, delamination.

By adding more nanoparticles to the melt (i.e., twice the AMNC2), the wear mechanism of AMNC4 has not been considerably changed, as seen in Figure 12c. A combination of abrasion and delamination mechanisms is still detectable. The addition of further nanoparticles to the melt in AMNC4 led to an increased viscosity and hence a decreased ultrasonic treatment performance. With the increase in the porosity amount, the bonding strength between the reinforcing agent and matrix is reduced; therefore, the cracks can be initiated and grown as seen in some parts of the abraded surface. Probably, these cracks have spread and, in the parts with weaker bonding, relatively large pieces have separated and created pits on the surface. Studying the flash, it can be seen that there is no noticeable change between the reinforced composite samples. In other words, having the nanoparticles in the metal matrix, the amount of plastic deformation has been reduced and the developed flash has, in all likelihood, regularly been separated from the surface during the wear test.

Considering the issue of insufficient bonding of nanoparticles in higher weight fractions and the possibility of agglomerations, abrasion mechanism is seen on a small part of the AMNC6 surface and looks like islands in the middle of the entire surface (see Figure 12d). In addition, delamination is also recognizable with higher thicknesses. Deep grooves are probably created by the hard silica particles separated from the abraded surface to facilitate three-body wear. The aforesaid hard particles could be either those particles weakly bonded to the base matrix or the agglomerated phase that has been fragmented by wear forces.

An increase in the hardness of the nanocomposite may reduce the materials transfer from the pin surface to the disc surface. In other words, nanoparticles and the hardened structure of the pin surface tend to partially diffuse into the disc, facilitating the iron transfer to the pin. In long distances and increasing temperature, oxygen could be absorbed to form iron oxide as a tribolayer [53,72]. In Figure 13a–d, the wear debris of monolithic and composite samples reinforced with different weight fractions, along with the EDS results, are presented (Figure 13e–h). Seen in Figure 13, the wear debris of the unreinforced samples is greater, irregular in shape, thicker, and their deformed shapes are more pronounced. This type of chipping mostly occurs during the plastic deformation of the material, being a feature of adhesive mechanism. The size of the debris in the AMNC2 sample is relatively smaller, and their thickness has also been reduced. Some plate-like chips are also observed in AMNC2. This type of chipping is usually seen when material is moved at the edges of the wear grooves. In the samples AMNC4 and AMNC6, a combination of the all kinds of chips with irregular shape, sheet-like, and flaky morphology is observed. Figure 13 shows the EDS result of monolithic and A356/SiO$_2$ nanocomposite samples; Fe and O elements are detected.

These elements, combined with the temperature rise during the wear test, provide the conditions for the formation of oxidized wear. The strength and weakness of this mechanism are demonstrated by the different percentages of iron observed in the EDS analysis. For example, as shown in Figure 13e, the percentage of iron transferred to the

surface of the pin is 3.55%, while this amount has reached about 28% in the AMNC2 sample. In samples with a higher weight fraction of nanosilica, the amount of transferred iron has been reduced by reducing the hardness. The amount of iron transferred from the disc surface to the pin surface is probably related to the hardness of the composite samples. In other words, by increasing the hardness of the samples, the penetration of the asperity of the pin surface to the disk increases and more iron is transferred to the pin. As the temperature rises, the conditions for the formation of an oxide layer are provided, preventing direct contact between metal and metal, and the amount of wear decreases [73,74].

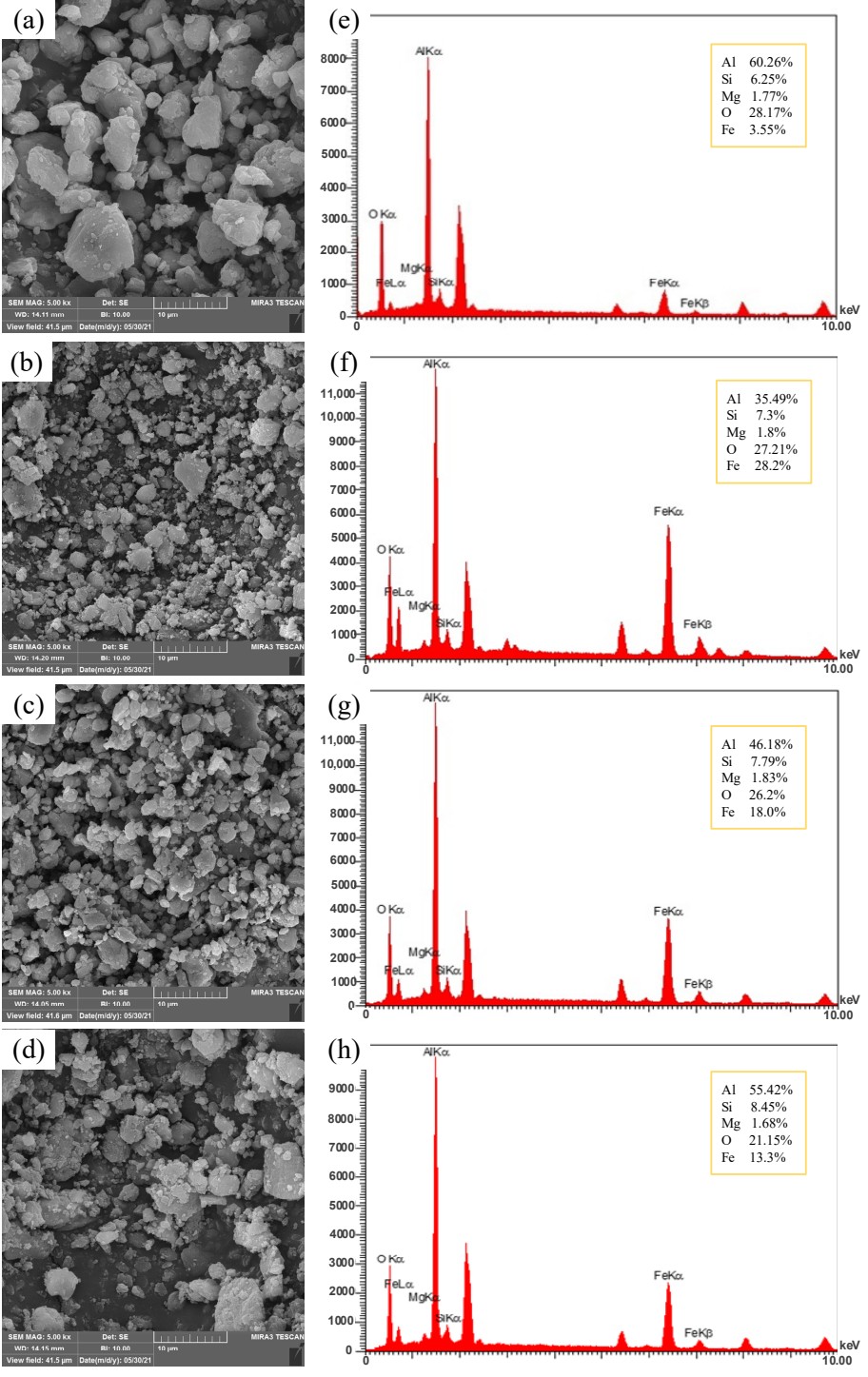

**Figure 13.** SEM image of debris: (**a**) unreinforced alloy, (**b**) AMNC2, (**c**) AMNC4, (**d**) AMNC6. The EDS of wear debris for the (**e**) unreinforced alloy, (**f**) AMNC2, (**g**) AMNC4, and (**h**) AMNC6.

## 4. Conclusions

The main conclusions of the present study are described as follows:

1.  A proper ultrasonic melt treatment caused a better dispersion of the nanosilica reinforcements in the A356 aluminum alloy and refined the microstructure of the fabricated composite materials. Adding nanoparticles with 0.125 wt.% resulted in better wear resistance due to the synergistic improvement in both the reinforcing role of the nanosilica particles and ultrasonic efficiency. Adding more than this amount (0.125 wt.%), however, resulted in a relatively coarse microstructure due to the increase in viscosity and thus a decrease in the efficiency of ultrasonic waves;

2.  Based on the hardness profile in the longitudinal direction of the cast ingot, the highest hardness is measured at the bottom and then at the top of the cast ingot. The hardness of all samples reinforced with nanosilica particles was higher than the base alloy. Among these, the largest increase is obtained from the A356/0.125% $SiO_2$ sample, exhibiting a 52% hardness increase;

3.  The pin-on-disc experiment revealed that all the composite samples have a lower weight loss rate compared to the monolithic material. Investigations showed that the nanocomposites have better wear resistance at higher forces; for example, the A356/0.125% $SiO_2$ sample under the force of 100 N has recorded an improvement in wear resistance of about 68%;

4.  As much as 50% improvement of the COF in the sample with the optimal wt.% (A356/0.125% $SiO_2$), and a reduction in the COF of all composite samples compared to the base alloy, is seen in this research. The COF in the composite samples did not change much with the increase in force, but the COF of the matrix alloy decreased with the addition of force;

5.  Examining the abraded surfaces revealed that the dominant wear mechanisms in the samples reinforced with nanosilica are mainly abrasion and delamination. Further, the amount of abrasion in the A356/0.125% $SiO_2$ composite sample is higher and the destruction in this sample has reached its lowest level. EDS investigation of the wear debris showed that the iron transfer from the disc surface to the pin surface has taken place with the addition of nanosilica, and the iron transfer increased with higher hardness.

**Author Contributions:** All authors contributed to the study conception and design. Material preparation, data collection, and analysis were performed by all authors. The first draft of the manuscript was written by A.G. and all authors commented on previous versions of the manuscript. All authors have read and agreed to the published version of the manuscript.

**Funding:** This research received no external funding.

**Data Availability Statement:** The datasets generated and/or analyzed during the current study are available in this manuscript and there are no other data that are not in this manuscript.

**Conflicts of Interest:** The authors declare no conflict of interest.

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
