# Peer review of "An Investigation on the Enhanced Wear Behavior of Ultrasonically Stirred Cast A356/SiO2np Nano-composites"

_crystals, doi:10.3390/cryst13050722_

Round 1

Reviewer 1 Report

In the present study is proposed a study on the enhanced wear behavior of ultrasonically stirred and cast A356/SiO2np nano-composites.

The topic of the paper is very interesting, the structure of the paper is well organized and well written, and it is easy to follow.

In the reviewer opinion, the present paper meets all requirements for the published in Crystals journal.

Minor comments:

1. Page 1, line 35

The meaning of CNT should be presented.

2. Page 3, table 2

In the units colon should be used superscript when appropriate (examples: square meters, cubic meters,…).

3. Page 4, Fig. 2 a)

Error in the text box: should be: ...transducer".

4. Page 5, Fig. 3a)

Text boxes should be improved.

5. Page 6, table 3

Capital letter in the title and in other parameters of the table.

6. Page 6, line 211

Detail why these names of the samples...

7. Page 8, Fig. 7

Correct in the vertical axis of fig a) the units to HV.

Correct in the horizontal axis of fig b) the legend, that is not a percentage.

8. Page 12, line 381

The number of this section is duplicated, should be 3.2.

The same comment in the next section number.

9. Page 17, Fig. 13

Legend of figures, specially from e) to h), should be improved.

Author Response

Dear Reviewer,

We appreciate you for your precious time in reviewing our paper and providing valuable comments. It was your valuable and insightful comments that led to
possible improvements in the current version. The authors have carefully considered the comments and tried our best to address every one of them. We hope the manuscript after careful revisions meet your standards. The authors welcome further constructive comments if any. 

Revised manuscript uploaded and changes are highlighted in green and some comments made in side tracks.

Sincerely,

Ahmad Ghahremani

Reviewer 2 Report

See my pdf file and yellow notes in them. They indicate problems in English, in subchapters numbering and some figure captions. These problems are not a key factor concerning the technical merit of the text. In general: I can agree with your manuscript to be published.

Se the above box.

Author Response

(The authors gave the same response as above.)
